# (*E*)-1-(Furan-2-yl)-(substituted phenyl)prop-2-en-1-one Derivatives as Tyrosinase Inhibitors and Melanogenesis Inhibition: An In Vitro and In Silico Study

**DOI:** 10.3390/molecules25225460

**Published:** 2020-11-21

**Authors:** Hee Jin Jung, Sang Gyun Noh, Il Young Ryu, Chaeun Park, Ji Young Lee, Pusoon Chun, Hyung Ryong Moon, Hae Young Chung

**Affiliations:** 1College of Pharmacy, Pusan National University, Busan 46241, Korea; hjjung2046@pusan.ac.kr (H.J.J.); rskrsk92@naver.com (S.G.N.); iy2355@naver.com (I.Y.R.); myceaun@nate.com (C.P.); sjn04028@gmail.com (J.Y.L.); 2College of Pharmacy and Inje Institute of Pharmaceutical Sciences and Research, Inje University, Gimhae, Gyeongnam 50834, Korea; pusoon@inje.ac.kr

**Keywords:** furan-chalcone, melanogenesis, molecular dynamics, tyrosinase inhibitor

## Abstract

A series of (*E*)-1-(furan-2-yl)prop-2-en-1-one derivatives (compounds **1**–**8**) were synthesized and evaluated for their mushroom tyrosinase inhibitory activity. Among these series, compound **8** (2,4-dihydroxy group bearing benzylidene) showed potent tyrosinase inhibitory activity, with respective IC_50_ values of 0.0433 µM and 0.28 µM for the monophenolase and diphenolase as substrates in comparison to kojic acid as standard compound 19.97 µM and 33.47 µM. Moreover, the enzyme kinetics of compound **8** were determined to be of the mixed inhibition type and inhibition constant (*K*_i_) values of 0.012 µM and 0.165 µM using the Lineweaver-Burk plot. Molecular docking results indicated that compound **8** can bind to the catalytic and allosteric sites 1 and 2 of tyrosinase to inhibit enzyme activity. The computational molecular dynamics analysis further revealed that compound **8** interacted with two residues in the tyrosinase active site pocket, such as ASN260 and MET280. In addition, compound **8** attenuated melanin synthesis and cellular tyrosinase activity, simulated by α-melanocyte-stimulating hormone and 1-methyl-3-isobutylxanthine. Compound **8** also decreased tyrosinase expressions in B16F10 cells. Based on in vitro and computational studies, we propose that compound **8** might be a worthy candidate for the development of an antipigmentation agent.

## 1. Introduction

Tyrosinase (EC 1.14.18.1) is a metalloenzyme group of polyphenol oxidases that is found in various organisms and exhibits a specific function in melanogenesis [1]. Tyrosinase promotes the transformation of l-tyrosine to l-3,4-dihydroxyphenylalanine (l-DOPA) (monophenolase activity) by hydroxylation, and then, l-DOPA is converted to DOPA-quinone (diphenolase activity) [2,3,4]. When l-tyrosine or l-DOPA are the substrates, the product of the reaction catalyzed by tyrosinase is dopaquinone, an intermediate in the melanin biosynthesis pathway [5].

Melanin, whose production is catalyzed by tyrosinase, exists in bacterial, fungi, plants and keratinocytes of the skin and hair of animals. It plays a vital role in protecting the skin from damage, such as premature aging, as well as in imparting protection to the eyes against ultraviolet radiation [6]. On the other hand, the overproduction and accumulation of pigment may lead to epidermal pigmentation and cause certain skin dermatological disorders, such as melasma, freckles, age spots, senile lentigines and one of the most dangerous skin cancers, malignant melanoma [7,8]. Kojic acid and hydroquinone, which were used as ingredients of skin-lightening cosmetics, have been reported to be unsafe for human skin at the concentrations required for their depigmenting effects [9]. For this reason, there is a need to discover new tyrosinase inhibitors that can be safely applied to the skin. 

Studies revealed that compounds with chalcone-based structures possess a wide variety of biological and pharmacological activities, such as anti-proliferation [10], anti-inflammation [11], anti-neurodegeneration [12,13,14], anti-cancer [15], anti-biotic [16], anti-bacterial [17], antioxidant [18] and anti-nociceptive [19] activities. Furthermore, previous investigations reported the application of chalcone derivatives [20,21,22] and keto- and carboxy-furan derivatives [23] as promising tyrosinase inhibitors. 

Recently, we investigated that chalcone hearing-substituted benzylidene thiophen (a heterocyclic compound with the formula C4H4S) derivatives showed remarkable anti-tyrosinase and anti-melanogenesis activities [24]. In light of the significant activity of chalcone framework derivatives in the attenuation of tyrosinase and melanin productions, which have also been associated with the development of skin pigmentation, these framework derivatives were further synthesized in our laboratory and evaluated with respect to their anti-tyrosinase properties. We here report the findings on the tyrosinase and melanogenesis activities of the chalcone-bearing furan (a heterocyclic organic compound).

In this regard, the objective of the present work was to identify the anti-tyrosinase and anti-melanogenesis activities of furan-chalcone derivatives using in vitro enzyme kinetics and computational studies, including molecular docking simulations and molecular dynamics, and α-melanocyte-stimulating hormone (α-MSH) and 3-isobutyl-1-methylxanthine (IBMX)-induced melanin contents, cellular tyrosinase activity and tyrosinase protein expression in B16F10 cells. Considering the results of these above experiments, we determined the prospect of (*E*)-1-(furan-2-yl)prop-2-en-1-one derivatives as skin pigmentation disease agents.

## 2. Results and Discussion

### 2.1. Procedure for the Synthesis of (E)-1-(Furan-2-yl)-3-(substituted phenyl)prop-2-en-1-one Derivatives **1**–**9**

The furan-2-yl-substituted benzaldehyde analogs **1**–**9** were shown in Scheme 1. For the synthesis of (*E*)-1-(furan-2-yl)-3-(substituted phenyl)prop-2-en-1-one derivatives **1**–**8**, 2-acetylfuran was reacted with the appropriate substituted benzaldehydes under acidic conditions using 1-M HCl acetic acid solution. The acidic aldol condensation reactions provided the desired compounds **1**–**6** as solids. However, the reaction of 2-acetylfuran with 3,5-dibromo-4-hydroxybenzaldehyde or 2,4-dihydroxybenzaldehyde under acidic conditions such as using 1-M HCl acetic acid solution did not give the desired products **7** and **8**. For the synthesis of the desired compounds **7** and **8**, a Claisen-Schmidt reaction was attempted under basic conditions. Fortunately, compound **7** was obtained directly by the Claisen-Schmidt reaction between 2-acetylfuran and 3,5-dibromo-4-hydroxybenzaldehyde using 1-M NaOH, and 2,4-dihydroxyphenyl compound **8** was prepared by the *O*-demethylation of compound **9**, which was produced via the Claisen-Schmidt reaction between 2-acetylfuran and 2,4-dimethoxybenzaldehyde using 1-M NaOH. The double-bond geometry of the final compounds was easily confirmed by the coupling constant (*J*) values (ca. 16.0 Hz) of the vinylic protons, indicating that the alkenes are (*E*) isomers. The structures of the final products were confirmed by HR-ESI-MS, ^1^H-NMR and ^13^C-NMR spectral data.

### 2.2. Mushroom Tyrosinase Inhibitory Activities of (E)-1-(Furan-2-yl)-(substituted phenyl)prop-2-en-1-one Derivatives (Compounds **1**–**8**)

In order to evaluate the inhibitory effects of synthesized furan chalcone derivatives on the enzyme activity, tested compounds were subjected to a mushroom tyrosinase inhibition assay with monophenolase (l-tyrosine) or diphenolase (l-3,4-dihydroxyphenyl-alanine; l-DOPA) as the substrates. Since kojic acid is well-known as a standard tyrosinase inhibitor [25,26], phthalic acid and cinnamic acid are potent mixed-type inhibitors of mushroom tyrosinase [27]. The inhibition results are tabulated in Table 1. Substitution of the furan chalcone with a 2,4-dihydroxyphenyl group (compound **8**) resulted in a potent tyrosinase inhibitory activity, with 50% inhibitory concentration (IC_50_) values of 0.0433 ± 0.0016 µM for l-tyrosine and 0.28 ± 0.01 µM for l-DOPA, which explains its excellent tyrosinase inhibitory activity compared to the positive control, kojic acid (19.97 ± 0.36 µM for l-tyrosine and 33.47 ± 0.05 µM for l-DOPA). These results are in agreement with previous studies, suggesting that the 2,4-dihydroxyphenyl group attached to the thiophene chalcones has an important role in determining their activity [28]. Compound **4** with a 3-hydroxy-4-methoxyl group on the phenyl ring exhibited significant inhibitory activity (13.61 ± 3.74 µM to 37.336 ± 4.89 µM) against mushroom tyrosinase. On the other hand, a 4-hydroxy group alone on the β-phenyl ring showed moderate tyrosinase inhibitory activity, while the insertion of an additional functional group into the three position of the β-phenyl ring dramatically diminished the inhibition (compounds **2**, **3** and **6**). Notably, compounds with a 3-hydroxy-4-methoxy group on the β-phenyl ring exerted stronger inhibitory activity than a 4-hydroxy-3-methoxy group (**3** vs. **4**). These results suggest that the number and type of the functional group on the β-phenyl ring of furan chalcone derivatives greatly affects the inhibitory activity of tyrosinase.

Notably, compound **8** exhibited dose-dependent inhibitory activities with both substrates. The concentration-dependent mushroom tyrosinase inhibitory graph of compound **8** shows a dose-dependent inhibition with both substrates, which demonstrated an initial dose-dependent activity followed by decreased enzyme activity starting at a concentration of 0.4 µM (Figure 1A,B). These results show that low doses of compound **8** showed a dose-dependent inhibition of monophenolase, whereas high doses of this compound significantly increased the enzyme activity. This indicates that the binding site of compound **8** might play an important role in monophenolase. 

Based on our earlier results, we reported that a potent tyrosinase inhibitor (*E*)-2,4-dihydroxyphenyl-(thiophen-2-yl)prop-2-en-1-one showed a higher antityrosinase activity, with IC_50_ values of 0.013 µM for monophenolase and 0.93 µM for diphenolase when compared to the other derivatives [24]. For these reasons, the present findings could be considered as a basis for further studies aiming at better defining the functional group of chalcone framework derivatives. According to our results, although the introduction and position of the functional group played an important role in structure-activity relationship studies against the tyrosinase enzyme, but it is also significant for their overall framework—at least, the tyrosinase inhibitory activity.

### 2.3. Enzyme Kinetics Mechanism Study

The tyrosinase inhibition kinetics of compound **8** were analyzed using Lineweaver-Burk plots. Lineweaver-Burk plots were performed with different concentrations of l-tyrosine (Figure 2A) and l-DOPA (Figure 2D) as substrates and various concentrations of compound **8**. In the Lineweaver-Burk plots, the lines for various concentrations of compound **8** were intersected on the left side, indicating mixed-type inhibition against tyrosinase (Figure 2A,D) in both substrates. Compound **8** showed a mix-type inhibition. In accordance with the mixed-type inhibition, inhibitors likely bind the enzyme and the enzyme-substrate complex [29,30]. To calculate *K*_i_, the slope and the intercept of the Lineweaver-Burk lines for concentrations of compound **8** were plotted as a function of the inhibitor concentration (Figure 2B,C for l-tyrosine and Figure 2E,F for l-DOPA). The intersections of the lines from the slope and intercept with the X-axis showed *K*_i_ values (12 nM and 165 nM) and α*K*_i_ values (101 nM and 505 nM) when l-tyrosine and l-DOPA were used as the substrates, respectively. The *K*_i_ value was considerably lower than that recently reported for kojic acid (9.23 µM) [21], which demonstrates that compound **8** is a potent tyrosinase inhibitor.

### 2.4. Molecular Docking Simulation of Compound **8** with Mushroom Tyrosinase

To examine whether compound **8** binds to the catalytic and allosteric sites 1 and 2 of tyrosinase and inhibits the enzyme as a result of the binding, an in silico docking simulation was carried out using AutoDock 4.1.2. software (The Scripps Research Institute, La Jolla, CA, USA). The 3D X-ray crystal structure of *Agaricus bisporus* (*A. bisporus*) tyrosinase was downloaded from the Protein Data Bank (PDB ID:2Y9X) for a docking simulation with compound **8** and kojic acid or cinnamic acid and phthalic acid, as mixed-type tyrosinase inhibitors [27,31]. Mushroom *A. bisporus* is used to its commercially available and high homology with a mammalian tyrosinase enzyme that renders it well-studied as a model for studies on melanogenesis [32].

A molecular docking study was performed with an emphasis on compound **8**, which showed the greatest tyrosinase inhibitory activity (Table 1 and Figure 3). The molecular docking models of compound **8** and kojic acid (a well-known competitive-type inhibitor) [33] in the catalytic site of tyrosinase are represented in Figure 3A. The tyrosinase-compound **8** inhibitor complex presented5.69-kcal/mol binding energy, including two hydrogen bonds with the ASN260 and MET280 residues of tyrosinase, and hydrophobic interactions were observed between compound **8** and the tyrosinase residues VAL248, MET257, PHE264, VAL283 and ALA286, which further stabilized the interaction in the catalytic site of mushroom tyrosinase (Table 2 and Figure 3B,E), and kojic acid binds to HIS263 and MET280 (Figure 3H). Moreover, the molecular docking of compound **8** and of phthalic acid and cinnamic acid were taken as reference ligands at allosteric sites 1 and 2, respectively [27]. As shown in Figure 3C,F, compound **8** exhibited five hydrophobic interactions residues with the LEU24, TYR140, PHE147, ILE217 and ALA221 residues of tyrosinase, and phthalic acid displayed interactions with the TRP136, ILE217, ALA221, PHE224 and LEU265 residues of tyrosinase at allosteric site 1 (Figure 3I). The corresponding ligand interaction of **8** in the allosteric site 2 of tyrosinase included three hydrogen bonds with the THR308, ASP312 and GLU356 residues and a hydrophobic interaction at the TYR311 residue (Figure 3D,G) and cinnamic acid binding with the LYS379 residue (Figure 3J). Furthermore, compound **8**–tyrosinase binding was found to exert binding energy in both allosteric sites 1 and 2 of tyrosinase (−4.52 and −5.72 kcal/mol, respectively), which indicated a high binding affinity with both allosteric sites (Table 2). Based on enzyme kinetic studies, compound **8** exerted a mixed-type inhibition; the binding ability of **8** with both the catalytic site and two allosteric sites 1 and 2 confirmed its mixed-type inhibition of tyrosinase. Therefore, compound **8** holds great promise for the treatment of pigmentation through tyrosinase inhibition. However, the details of the specificity/selectivity for human tyrosinase homologs and 3D docking simulations are still unknown and will be a challenge that needs to be confirmed in the future.

### 2.5. Molecular Dynamics (MD) Simulation Analyses

A MD simulation is generally used to analyze the conformational changes and stability of biomolecules over a period of time, complementing and verifying the docking results [34]. To estimate the mushroom tyrosinase flexibility and overall stability of docking complexes, we carried them out using the Gromacs 5.1.2 software package. The residual deviations and fluctuations of compound **8** and kojic acid were determined by the root mean square deviation and fluctuation (RMSD/F) graphs generated by Gnuplot software.

With these objectives, we verified the RMSD of the carbon α-atoms with a tyrosinase bound to compound **8**, and the results showed that the structure was first rearranged and then stabilized (Figure 4A). The RMSD for the tyrosinase–compound **8** was eventually stable at around 0.2 nm and 0.5 nm during simulations. Furthermore, in order to verify the movement of amino acid residues binding to tyrosinase of compound **8** during the MD simulation, we plotted the RMSFs for carbon α-atoms of all residues. The RMSF plots of the tyrosinase–compound **8** complex (purple) and tyrosinase–kojic acid complex (green) were generated (Figure 4B). We found that the RMSF curve was similar to that of the tyrosinase–compound **8** complex or tyrosinase–kojic acid complex, despite the difference in RMSF values for some residues, such as the 208–243 residue numbers, indicating that compound **8** binds tightly with the active site of tyrosinase.

Based on the results of MD simulations, hydrogen residues of compound **8** with possible interactions with tyrosinase were selected. As mentioned before, ASN260 and MET280 were identified as the hydrogen-bonded residues within compound **8** (Figure 3). Therefore, we confirmed the distance between tyrosinase and the residue of ASN260 in compound **8** (Figure 4C). After that, the distances between tyrosinase–compound **8** were represented at around 0.2 nm and 0.3 nm after stabilization, proving that tyrosinase interacts with the hydrogen bonds of compound **8**. On the other hand, the distance between tyrosinase–compound **8** was displayed at around 0.15 nm to 2 nm in the residue of MET280, while the distance between tyrosinase–kojic acid was around 0.4 nm to 0.5 nm, indicating that the binding between compound **8** and tyrosinase was a more relatively stable binding than as compared to that between kojic acid and tyrosinase (Figure 4D). Therefore, these computational analyses results are in agreement with the enzymatic activity presented in this study, demonstrating that compound **8** is capable of strongly inhibiting the tyrosinase enzyme.

### 2.6. Cell Viability of Compound **8** in B16F10 Cells

The effect of compound **8** on the viability of B16F10 cells was assessed with an Ez-Cytox assay method. B16F10 cells were treated with compound **8** at concentrations ranging from 1–20 µM, and viability was determined after 24 h (Figure 5A) or 48 h (Figure 5B). Concentrations up to 20 µM of the compound showed no cytotoxicity on B16F10 cells. Hence, up to 20 µM of compound **8** was used for all subsequent experiments, such as melanin contents, cellular tyrosinase activity and tyrosinase protein expression in B16F10 cells.

### 2.7. Melanin Contents Measurement of Compound **8**

Although compound **8** was active against tyrosinase primarily by inhibition of the melanogenic pathway, we were interested in the inhibition of hyperpigmentation disorders. To investigate the effect of compound **8** on anti-melanogenesis, B16F10 cells were treated with various concentrations (1, 5, 10 and 20 µM) of compound **8** for 48 h, and the melanin content was measured from the extracellular and intracellular compartments. Treatment with the α-melanocyte-stimulating hormone (α-MSH) and 3-isobutyl-1-methylxanthine (IBMX) significantly increased the extracellular and intracellular melanin contents by approximately 210% (Figure 6A) and 280% (Figure 6B), respectively, compared with the untreated group. Hence, compound **8** decreased the α-MSH and IBMX-mediated increases in the extracellular and intracellular melanin contents in a concentration-dependent manner. As depicted in Figure 6, we found that compound **8** remarkably decreased the extracellular and intracellular melanin. These data indicated that compound **8** is potentially a promising anti-melanogenic agent.

Although numerous reports have only determined the intracellular melanin in the process of effective compounds that are suspected to regulate melanogenesis, the results of our determination show that most of the melanin synthesized by the B16F10 cells was released outside the cells. Therefore, to elucidate the regulation of melanogenesis by compound **8**, it is necessary to determine not only the intracellular but also the extracellular activity of this compound.

### 2.8. Cellular Tyrosinase Activities and Tyrosinase Protein Levels of Compound **8**

Tyrosinase is a copper-dependent enzyme that catalyzes the conversion of l-tyrosine to l-DOPA, the rate-limiting step in melanin biosynthesis [1,35]. Therefore, we evaluated whether compound **8** inhibits tyrosinase activity in B16F10 cells [36]. To determine the cellular tyrosinase activity and expression of tyrosinase, B16F10 cells were incubated for 48 h with various concentrations (1, 5, 10 and 20 µM) of compound **8** in the absence or presence of α-MSH (1 µM) and IBMX (200 µM). As shown in Figure 7, both the cellular tyrosinase activity and expression of tyrosinase were dramatically increased by α-MSH and IBMX stimulation, whereas these upregulated expression levels and inhibition activity of tyrosinase were decreased by the treatment with compound **8** in a concentration-dependent manner (Figure 7). The inhibitory effect of the compound **8** was even much stronger than that of kojic acid, the positive control (Figure 7A). All these results suggest that compound **8** reduces melanogenesis by decreasing the cellular tyrosinase activity and expression of tyrosinase in B16F10 cells. Further studies are required to explain the different results between the downregulation of the tyrosinase expression and the inhibition of cellular tyrosinase and melanin formation.

## 3. Material and Methods

### 3.1. Chemicals and Reagents

Mushroom tyrosinase (EC 1.14.18.1), α-melanocyte-stimulating hormone (α-MSH), 3-isobutyl-1-methylxanthine (IBMX), l-tyrosine, l-3,4-dihydroxyphenylalanine (l-DOPA), dimethyl sulfoxide (DMSO), kojic acid, phthalic acid and *trans*-cinnamic acid were acquired from Sigma-Aldrich (St. Louis, MO, USA). Dulbecco’s modified Eagle’s medium (DMEM), fetal bovine serum (FBS), penicillin-streptomycin, 0.25% trypsin-ethylenediaminetetraacetic acid (EDTA) and amphotericin were purchased from WELGENE Inc. (Gyeongsan-si, South Korea). All other reagents were purchased from Sigma-Aldrich. All other chemicals and solvents were purchased from Merck (Frnakfurt Str., Darmstadt, Germany), Fluka (St. Louis, Mo, USA) and Sigma-Aldrich unless otherwise stated.

### 3.2. General Experimental Procedures

All reagents were obtained commercially and used without further purification. Thin-layer chromatography (TLC) and column chromatography were conducted on Merck precoated 60F245 plates and MP Silica 40–63, 60 Å, respectively. High-resolution (HR) mass spectroscopy data were obtained on an Agilent Accurate Mass Q-TOF (quadruple time-of-flight) liquid chromatography (LC) mass spectrometer (Agilent, Santa Clara, CA, USA) in ESI-positive mode. Nuclear magnetic resonance (NMR) spectra were recorded on a Varian Unity INOVA 400 spectrometer or a Varian Unity AS500 spectrometer (Agilent Technologies, Santa Clara, CA, USA) for ^1^H NMR (400 and 500 MHz) and for ^13^C NMR (100 MHz). DMSO-*d_6_*, CD_3_OD and CDCl_3_ were used as solvents for NMR samples. The coupling constant (*J*) and chemical shift values were measured in hertz (Hz) and parts per million (ppm), respectively. The abbreviations used in the analysis of ^1^H NMR data are follows: s (singlet), brs (broad singlet), d (doublet), dd (doublet of doublets), t (triplet), td (triplet of doublets), q (quartet) and m (multiplet).

#### 3.2.1. General Procedure for the Preparation of (*E*)-1-(Furan-2-yl)-3-(substituted phenyl)prop-2-en-1-one Derivatives **1**–**6**

A solution of 2-acetylfuran (100 mg, 0.91 mmol) and appropriately substituted benzaldehyde (1.0 equiv.) in 1-M HCl acetic acid (2 to 3 mL) was stirred at ambient temperature for 1–3 days. Work-up for compounds **1**–**5**: The reaction mixture was neutralized to pH 7 using aqueous NaHCO_3_ solution and was partitioned between ethyl acetate or methylene chloride and water. The organic layer was dried over anhydrous Na_2_SO_4_, filtered and evaporated under reduced pressure. The crude solid was recrystallized in methanol and water to give (*E*)-1-(furan-2-yl)-3-(substituted phenyl)prop-2-en-1-one derivatives **1**–**5** as solids. Work-up for compound **6**: After adding water to the reaction mixture, the resulting precipitate was filtered to generate compound **6** (22.3 mg, 8.4%) as a solid. Structure characterization (^1^H- and ^13^C-NMR and ESI-MS data of compounds **1**–**8** is provided in the Appendix A.

##### (*E*)-1-(Furan-2-yl)-3-(4-hydroxyphenyl)prop-2-en-1-one (**1**)

Dark-yellow solid; 41.9 mg (21.5%); ^1^H MMR (400 MHz, DMSO-*d*_6_) *δ* 10.07 (s, 1H, OH), 7.99 (s, 1H, 5′-H), 7.69 (d, 1H, *J* = 3.6 Hz, 3′-H), 7.66 (d, 2H, *J* = 8.0 Hz, 2′′-H, 6′′-H), 7.63 (d, 1H, *J* =15.6 Hz, 3-H), 7.45 (d, 1H, *J* = 15.6 Hz, 2-H), 6.80 (d, 2H, *J* = 8.0 Hz, 3′′-H, 5′′-H), 6.73 (m, 1H, 4′-H); ^13^C-NMR (100 MHz, DMSO-*d*_6_) *δ* 177.4, 160.8, 153.8, 148.6, 143.9, 131.6, 126.2, 119.3, 119.0, 116.5, 113.3; HR-MS (ESI+) *m/z* C_13_H_11_O_3_ (M + H)^+^ calcd 215.0703, obsd 215.0701.

##### (*E*)-3-(3,4-Dihydroxyphenyl)-1-(furan-2-yl)prop-2-en-1-one (**2**)

Greenish-yellow solid; 12.0 mg (5.7%); ^1^H MMR (400 MHz, DMSO-*d*_6_) *δ* 9.36 (brs, 2H, 3′′-OH, 4′′-OH), 7.99 (s, 1H, 5′-H), 7.67 (d, 1H, *J* = 3.6 Hz, 3′-H), 7.55 (d, 1H, *J* = 15.6 Hz, 3-H), 7.35 (d, 1H, *J* = 15.6 Hz, 2-H), 7.19 (s, 1H, 2′′-H), 7.11 (d, 1H, *J* = 8.4 Hz, 6′′-H), 6.76 (d, 1H, *J* = 8.4 Hz, 5′′-H), 6.72 (m, 1H, 4′-H); ^13^C-NMR (100 MHz, DMSO-*d*_6_) *δ* 177.4, 153.8, 149.5, 148.5, 146.3, 144.4, 126.7, 122.8, 119.2, 118.9, 116.4, 116.0, 113.3; HRMS (ESI+) *m/z* C_13_H_11_O_4_ (M + H)^+^ calcd 231.0652, obsd 231.0643.

##### (*E*)-1-(Furan-2-yl)-3-(4-hydroxy-3-methoxyphenyl)prop-2-en-1-one (**3**)

Greenish-yellow solid; 17.5 mg (8.6%); ^1^H MMR (400 MHz, DMSO-*d*_6_) *δ* 9.67 (s, 1H, OH), 8.00 (d, 1H, *J* = 2.0 Hz, 5′-H), 7.71 (d, 1H, *J* = 3.6 Hz, 3′-H), 7.63 (d, 1H, *J* = 15.6 Hz, 3-H), 7.47 (d, 1H, *J* = 15.6 Hz, 2-H), 7.41 (d, 1H, *J* = 2.4 Hz, 2′′-H), 7.23 (dd, 1H, *J* = 8.4, 2.4 Hz, 6′′-H), 6.80 (d, 1H, *J* = 8.4 Hz, 5′′-H), 6.74 (dd, 1H, *J* = 3.6, 2.0 Hz, 4′-H), 3.82 (s, 3H, OCH_3_); ^13^C-NMR (100 MHz, DMSO-*d*_6_) *δ* 177.4, 153.8, 150.4, 148.6, 148.6, 144.3, 126.7, 124.6, 119.4, 119.2, 116.3, 113.3, 112.4, 56.5; HRMS (ESI+) *m/z* C_14_H_13_O_4_ (M + H)^+^ calcd 245.0808, obsd 245.0806.

##### (*E*)-1-(Furan-2-yl)-3-(3-hydroxy-4-methoxyphenyl)prop-2-en-1-one (4)

Khaki solid; 40.6 mg (18.3%); ^1^H MMR (400 MHz, DMSO-*d*_6_) *δ* 9.16 (s, 1H, OH), 8.00 (s, 1H, 5′-H), 7.72 (d, 1H, *J* = 3.6 Hz, 3′-H) 7.58 (d, 1H, *J* = 15.6 Hz, 3-H), 7.42 (d, 1H, *J* = 15.6 Hz, 2-H), 7.25 (s, 1H, 2′′-H), 7.22 (d, 1H, *J* = 8.0 Hz, 6′′-H), 6.96 (d, 1H, *J* = 8.0 Hz, 5′′-H), 6.73 (m, 1H, 4′-H), 3.80 (s, 3H, OCH_3_); ^13^C-NMR (100 MHz, DMSO-*d*_6_) *δ* 177.3, 153.8, 151.0, 148.7, 147.3, 143.9, 128.0, 122.8, 119.9, 119.6, 115.3, 113.3, 112.6, 56.3; HRMS (ESI+) *m/z* C_14_H_13_O_4_ (M + H)^+^ calcd 245.0808, obsd 245.0802.

##### (*E*)-1-(Furan-2-yl)-3-(4-hydroxy-3,5-dimethoxyphenyl)prop-2-en-1-one (**5**)

Yellow solid; 20.3 mg (8.6%); ^1^H MMR (400 MHz, DMSO-*d*_6_) *δ* 9.05 (s, 1H, OH), 8.01 (s, 1H, 5′-H), 7.30 (d, 1H, *J* = 3.6 Hz, 3′-H), 7.64 (d, 1H, *J* = 15.6 Hz, 3-H), 7.50 (d, 1H, *J* = 15.6 Hz, 2-H), 7.12 (s, 2H, 2′′-H, 6′′-H), 6.75 (m, 1H, 4′-H), 3.81 (s, 6H, 2 × OCH_3_); ^13^C-NMR (100 MHz, DMSO-*d*_6_) *δ* 177.4, 153.8, 148.7, 148.7, 144.7, 139.5, 125.4, 119.6, 119.5, 113.2, 107.5, 56.8; HRMS (ESI+) *m/z* C_15_H_15_O_5_ (M + H)^+^ calcd 275.0914, obsd 275.0906.

##### (*E*)-3-(3-Bromo-4-hydroxyphenyl)-1-(furan-2-yl)prop-2-en-1-one (**6**)

Khaki solid; 22.3 mg (8.4%); ^1^H MMR (400 MHz, DMSO-*d*_6_) *δ* 10.88 (s, 1H, OH), 8.07 (d, 1H, *J* = 2.0 Hz, 2′′-H), 8.00 (s, 1H, 5′-H), 7.77 (d, 1H, *J* = 3.2 Hz, 3′-H), 7.62 (dd, 1H, *J* = 8.4, 2.0 Hz, 6′′-H), 7.59 (d, 1H, *J* = 15.6 Hz, 3-H), 7.52 (d, 1H, *J* = 15.6 Hz, 2-H), 6.96 (d, 1H, *J* = 8.4 Hz, 5′′-H), 6.74 (m, 1H, 4′-H); ^13^C-NMR (100 MHz, DMSO- *d*_6_) *δ* 177.2, 157.0, 153.7, 148.9, 142.3, 133.8, 130.8, 128.0, 120.6, 119.9, 117.1, 113.3, 110.8; HRMS (ESI+) *m/z* C_13_H_10_BrO_3_ (M + H)^+^ calcd 292.9808, obsd 292.9802, C_13_H_10_BrO_3_ (M + 2+H)^+^ calcd 294.9789, obsd 294.9787.

#### 3.2.2. Synthesis of Compound **7**

To a stirred solution of 2-acetylfuran (100 mg, 0.91 mmol) in methanol (3 mL) were added 1-M NaOH (1.8 mL, 1.8 mmol) and 3,5-dibromo-4-hydroxybenzaldehyde (254 mg, 0.91 mmol), and the reaction mixture was stirred at ambient temperature for 22 h. A 2-M HCl solution was added to the reaction mixture to adjust pH 7, and the generated precipitate was filtered and washed with water to provide compound **7** as a solid. 

##### (*E*)-3-(3,5-Dibromo-4-hydroxyphenyl)-1-(furan-2-yl)prop-2-en-1-one (**7**)

Yellowish-ivory solid; 82.9 mg (24.5%); ^1^H MMR (500 MHz, DMSO-*d*_6_) *δ* 10.51 (brs, 1H, OH), 8.11 (s, 2H, 2′′-H, 6′′-H), 8.04 (d, 1H, *J* = 1.5 Hz, 5′-H), 7.87 (d, 1H, *J* = 3.5 Hz, 3′-H), 7.65 (d, 1H, *J* = 16.0 Hz, 3-H), 7.59 (d, 1H, *J* = 16.0 Hz, 2-H), 6.77 (dd, 1H, *J* = 3.5, 1.5 Hz, 4′-H); ^13^C-NMR (100 MHz, DMSO-*d*_6_) *δ* 177.0, 153.6, 153.2, 149.2, 140.7, 133.3, 129.8, 122.4, 120.6, 113.3, 112.8; HRMS (ESI+) *m/z* C_13_H_9_Br_2_O_3_ (M + H)^+^ calcd 370.8913, obsd 370.8900, C_13_H_9_Br_2_O_3_ (M + 2 + H)^+^ calcd 372.8893, obsd 372.8880, C_13_H_9_Br_2_O_3_ (M + 4 + H)+ calcd 374.8874, obsd 374.8861.

#### 3.2.3. Synthesis of Compound **8**

To a stirred solution of compound **9** (50 mg, 0.19 mmol) in methylene chloride (1 mL) was added dropwise 1-M BBr_3_ methylene chloride solution (1.93 mL, 1.93 mmol) at 0 °C, and the reaction mixture was stirred at ambient temperature for 1.5 h. Five percent aqueous Na_2_HPO_4_ solution was added to the reaction mixture to adjust pH 6, and the reaction mixture was partitioned between ethyl acetate and water. The organic layer was dried over anhydrous MgSO_4_, filtered and evaporated under reduced pressure. The resulting residue was purified by a silica gel column chromatography using methylene chloride and methanol (20:1) as an eluent to give a sticky oil. For further purification, the sticky oil was dissolved in a small quantity of ethyl acetate, and then, hexane was added to the reaction mixture. The precipitate generated was filtered to afford compound **8** (10 mg, 22.4%) as a yellow solid.

##### (*E*)-3-(2,4-Dihydroxyphenyl)-1-(furan-2-yl)prop-2-en-1-one (**8**)

Yellow solid; ^1^H MMR (400 MHz, DMSO-*d*_6_) *δ* 10.18 (s, 1H, OH), 9.94 (s, 1H, OH), 7.96 (dd, 1H, *J* = 1.6, 0.8 Hz, 5′-H), 7.90 (d, 1H, *J* = 15.6 Hz, 3-H), 7.58 (d, 1H, *J* = 8.4 Hz, 6′′-H), 7.54 (dd, 1H, *J* = 3.6, 0.8 Hz, 3′-H), 7.40 (d, 1H, *J* = 15.6 Hz, 2-H), 6.69 (dd, 1H, *J* = 3.6, 1.6 Hz, 4′-H), 6.33 (d, 1H, *J* = 2.4 Hz, 3′′-H), 6.26 (dd, 1H, *J* = 8.4, 2.4 Hz, 5′′-H); ^13^C-NMR (100 MHz, DMSO-*d*_6_) *δ* 177.8, 162.1, 159.8, 154.1, 148.1, 139.5, 130.9, 118.4, 117.5, 113.7, 113.2, 108.6, 103.1; LRMS (ESI-) *m*/*z* 229 (M − H)^−^.

#### 3.2.4. Synthesis of Compound **9**

To a stirred solution of 2-acetylfuran (300 mg, 2.72 mmol) and 2,4-dimethoxybenzaldehyde (412 mg, 2.48 mmol) in ethanol (3 mL) was added dropwise 1-M NaOH aqueous solution (1.09 mL, 1.09 mmol), and the reaction mixture was stirred at ambient temperature for 4 h. After the addition of water, the reaction mixture was neutralized with 1-M HCl aqueous solution. The reaction mixture was partitioned between ethyl acetate and water, and the organic layer was dried over anhydrous MgSO_4_, filtered and evaporated under reduced pressure. The resultant residue was purified by a silica gel column chromatography using methylene chloride as an eluent to give compound **9** (498 mg, 77.8%).

##### (*E*)-3-(2,4-Dimethoxyphenyl)-1-(furan-2-yl)prop-2-en-1-one (**9**)

^1^H MMR (400 MHz, CDCl_3_) *δ* 8.12 (d, 1H, *J* = 15.5 Hz, 3-H), 7.62 (s, 1H, 5′-H), 7.58 (d, 1H, *J* = 8.0 Hz, 6′′-H), 7.44 (d, 1H, *J* = 15.5 Hz, 2-H), 7.27 (d, 1H, *J* = 3.5 Hz, 3′-H), 6.56 (m, 1H, 4′-H), 6.53 (d, 1H, *J* = 8.0 Hz, 5′′-H), 6.47 (s, 1H, 3′′-H), 3.90 (s, 3H, OCH_3_), 3.85 (s, 3H, OCH_3_); ^13^C-NMR (100 MHz, CDCl_3_) *δ* 178.9, 163.3, 160.7, 154.3, 146.3, 139.8, 131.2, 119.6, 117.2, 117.0, 112.5, 105.6, 98.6, 55.7, 55.7.

### 3.3. Mushroom Tyrosinase Inhibition Assay

Mushroom tyrosinase inhibition was determined following our previously reported methods [37]. All experiments were done in triplicate. Kojic acid, phthalic acid and cinnamic acid were used as standard compounds. The measurement was completed in triplicate for each concentration and averaged. The IC_50_ values were determined by interpolation of the dose-inhibition % curves.

### 3.4. Enzyme Kinetics Analysis of the Inhibition of Tyrosinase

The inhibition mode and inhibition constant (*K*_i_) and α*K*_i_ for mushroom tyrosinase inhibition were calculated from Lineweaver-Burk plots [29,38]. Kinetic parameters were obtained for various concentrations of the substrates (l-tyrosine and l-DOPA) and inhibitor (compound **8**). For Lineweaver–Burk double-reciprocal plots (a plot of 1/enzyme velocity (1/*V*) vs. 1/substrate concentration (1/[*S*])), the inhibition type was determined using various concentrations of l-tyrosine (0.5, 1, 2, 4 and 8 mM) or l-DOPA (0.5, 1, 2, 4 and 8 mM) as substrates in the presence of different concentrations of compound **8** (0, 20, 40 and 80 nM for l-tyrosine and 0, 125, 250 and 500 nM for l-DOPA). Thus, the type of enzyme inhibition and *K*_i_ and α*K*_i_ were determined by interpretation of the Lineweaver-Burk plots.

### 3.5. Computational Study

#### 3.5.1. Docking on the Mushroom Tyrosinase

Molecular docking analysis was carried out by a previously reported procedure [37]. The crystallographic structure of tyrosinase elucidated from the fungus *Agaricus bisporus* (PDB ID:2Y9X) [39]. The structures of compound **8** were converted to 3D structures by Marvin Sketch (v17.1.30, ChemAxon, Budapest, Hungary). AutoDock 4.2 was used for docking simulations, and grid maps were generated in the Autogrid program. The docking protocol for rigid and flexible ligand docking was composed of 10 independent genetic algorithms.

#### 3.5.2. Molecular Dynamics (MD) Analyses

The Gromacs 5.1.2 program was utilized to perform molecular dynamics simulations of the tyrosinase–compound **8** or tyrosinase–kojic acid (control) complex structures. Tyrosinase molecular force field parameters were written in Gromos53a6 force field format, and compound **8** or kojic acid molecular force field parameters were derived from Automated Topology Builder (ATB, https://atb.uq.edu.au/index.py) written in Gromos54a7 force field format, which is converted into Gromacs format data. At the beginning stage, energy minimization was executed using a steep descent method of 50,000 steps to have a stable conformation. After minimization, canonical ensembles (NVT) and isobar isothermal ensembles (NPT) were performed, respectively, with a constant temperature of 300 K for 100 ps for NVT, followed by a constant temperature of 300 K and a constant pressure of 1 atm per 100 ps for NPT. The production MD runs were then performed for 10 ns, keeping the temperature at 300 K and the pressure at 1 bar. The root mean square deviation (RMSD), root mean square fluctuation (RMSF) and distance between tyrosinase and compound **8** or kojic acid were calculated after the runs. The resulting graphics for these parameters were designed using the Gnuplot program.

### 3.6. Bioactivity

#### 3.6.1. Cell Culture and Cell Viability

The mouse melanocyte (B16F10) cell line was maintained in culture flasks containing Dulbecco’s modified Eagle’s medium (DMEM). The media was supplemented with 1% antibiotics and 10% heat-inactivated fetal bovine serum. After the cells formed a confluent monolayer, they were sub-cultured every two to three days. For the cell viability assays, cells were seeded in a 96-well plate at a density of 1 × 10^4^ cells/well and allowed to attach overnight. The cells were exposed to the indicated concentrations of compound **8** for 24 h or 48 h, and the cell viability was measured using the Ez-Cytox cell viability assay kit (Daeil Lab Service, Seoul, Korea) according to the manufacturer’s recommendations. Next, 10-µL Ez-Cytox (tetrazolium salts) was added to the medium, and the cells were incubated at 37 °C for 2 h. Control cells were treated with 0.1% DMSO, as DMSO exhibited no cytotoxicity at this concentration in the assay. The maximum concentration of vehicle (DMSO) in the culture media was adjusted to 0.1% (*v*/*v*). The absorbance was measured at 450 nm on a microplate reader (Tecan, Sunrise, Austria).

#### 3.6.2. Extracellular and Intracellular Melanin Contents Assay

The effect of an inhibitor on the melanin content in B16F10 cells was investigated according to the previous protocol [40]. B16F10 cells were cultured at a density of 1 × 10^4^ cells/mL in a 6-well plate, incubated for 24 h and then stimulated with α-MSH (1 µM) and IBMX (200 µM) in the presence or absence of indicated inhibitor concentrations or kojic acid for 48 h. Then, the culture media and the cell pellet were collected. The culture media was directly measured at 405 nm for extracellular melanin content. For intracellular melanin content, the cell pellets were washed with ice-cold PBS and dissolved in 100 µL of 1-M NaOH at 60 °C for 60 min. Then, the absorbance was measured at 405 nm on a microplate reader (Tecan, Sunrise, Switzerland). All results were normalized to the total protein concentration of the cell pellet using a Bicinchoninic Acid Assay kit (Thermo Fisher Scientific, Inc., Waltham, MA, USA). Kojic acid was used as a positive control.

#### 3.6.3. Cellular Tyrosinase Activity

B16F10 cells were seeded at 1 × 10^4^ cells/mL in a 6-well plate, incubated for 24 h and then stimulated with α-MSH (1 µM) and IBMX (200 µM) in the presence or absence of an inhibitor or kojic acid for 48 h. After treatment, the cells were washed with cold PBS and lysed by the addition of PBS containing 1% Triton X-100 at −80 °C for 1 h. The lysates were clarified by centrifugation at 8000 rpm for 15 min at 4 °C; then, 80 µL of cell lysate was added to 20 µL of l-DOPA (2 mg/mL in 50-mM phosphate buffer). The mixture was incubated for 10 min at 37 °C, and an absorbance at 492 nm of the reaction mixture was recorded [37]. Kojic acid was used as a positive control. 

#### 3.6.4. Western Blot Analysis

To determine the amount of protein expression, a Western blot analysis was performed. After treatment, B16F10 melanoma cells were lysed with the RIPA lysis buffer (Biosesang, Seongnam, Korea). Whole cell lysates were separated by 9% SDS-PAGE and transferred onto a polyvinylidene fluoride (PVDF) membrane (Millipore, Burlington, MA, USA). After blocking with 5% skimmed milk in PBS containing 0.1% Tween 20, the membranes were probed with specific primary antibodies overnight at 4 °C and then further incubated with horseradish peroxidase-conjugated secondary antibodies. Bound antibodies were detected by chemiluminescence using an ECL Prime Western Blotting Detection Reagent kit (GE Healthcare, Pittsburgh, PA, USA) following the manufacturer’s instructions. Antibodies for tyrosinase and β-actin were purchased from Santa Cruz Biotechnology (Dallas, TX, USA). 

### 3.7. Statistical Analysis

All data are presented as the mean ± the standard error of the median (SEM). Significant differences between groups were determined using an unpaired one-way ANOVA with Bonferroni correction. Values of ^***^
*p* < 0.001, ^**^
*p* < 0.01 and ^*^
*p* < 0.05 were considered to indicate statistical significances. The results shown in each of the figures are representative of at least three independent experiments.

## 4. Conclusions

To the best of our knowledge, we defined for the first time a series of furan (-substituted phenyl) chalcone derivatives (compounds **1**–**8**) with corresponding diverse functional groups as potent tyrosinase inhibitors and anti-hyper-pigmenting properties. An enzyme inhibitory assay revealed the importance of furan-chalcone bearing a 2,4-dihydroxyphenyl group (compound **8**) in exerting this anti-tyrosinase activity as a highly selective nanomolar tyrosinase inhibitor (IC_50_ = 43.3 nM for l-tyrosine and 280 nM for l-DOPA). Compound **8**, in particular, exhibited potent tyrosinase inhibitory activity kinetic studies, which revealed that compound **8** exhibited mixed-type inhibition against mushroom tyrosinase with the substrates l-tyrosine and l-DOPA. Moreover, computational studies indicated the binding modes of compound **8** with the amino acids of the active centers of tyrosinase’s catalytic and allosteric sites. Furthermore, compound **8** was found to be noncytotoxic in B16F10 cells and significantly suppressed melanin contents and intracellular tyrosinase activity, as well as the expression of tyrosinase. From the above results, compound **8** can serve as a structural template for designing inhibitors of the tyrosinase enzyme, which might provide new ideas for designing therapeutic agents against pigmentation diseases.

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
