# Peer review of "(E)-1-(Furan-2-yl)-(substituted phenyl)prop-2-en-1-one Derivatives as Tyrosinase Inhibitors and Melanogenesis Inhibition: An In Vitro and In Silico Study"

_molecules, 2020, doi:10.3390/molecules25225460_

Round 1

Reviewer 1 Report

The manuscript was clearly written except that there are places where grammar needs to be improved (examples are line 122, and line 256). The findings are new and exciting in that a furan derivative of benzaldehyde known as compound 8 could be useful as anti-pigmentation agent.

Author Response

Reviewer 1

The manuscript was clearly written except that there are places where grammar needs to be improved (examples are line 122, and line 256). The findings are new and exciting in that a furan derivative of benzaldehyde known as compound 8 could be useful as anti-pigmentation agent.

Answer: Thank you for your attention. In accordance, we revised these sentence in the Results and Discussion sections in line122 and 256 and noted with red color.

Reviewer 2 Report

Comments to the authors:

In the paper, Hee et al. reported  (E)-1-(Furan-2-yl)-(substituted phenyl)prop-2-en-1 2 one derivatives as tyrosinase inhibitors and investigated the effect on melanogenesis inhibition. Computational studies including docking and MD simulation revealed several key interactions with the enzyme. The authors demonstrate the non-cytotoxic effect and significant suppression of melanin contents and intracellular tyrosinase activity as well as the expression of tyrosinase.

Some comments below:

Major points:                                                                                                                   

  1. The compound contains a reactive Michael acceptor. Any possibility of covalent inhibition? Is there any time-dependent inhibition investigated in biochemical assays? The current scaffold could be promiscuous with a deluge of off-targets in the human proteome. The authors should explain the discrepancy between biochemical activity and cellular activity.

  1. The authors investigated MOA by molecular docking. Is that specific interaction that could explain the observed SAR? Without experimental validation like mutation, the docking for the allosteric site is not convincing enough.

  1. The figures look great but typo errors in the manuscript should be carefully checked. Also, Table 1 should be well-organized. Currently, the layout makes it unpleasant for reading. Texts in Figure 3D-I are difficult to see.

Author Response

Reviewer 2

In the paper, Hee et al. reported (E)-1-(Furan-2-yl)-(substituted phenyl)prop-2-en-1-2 one derivatives as tyrosinase inhibitors and investigated the effect on melanogenesis inhibition. Computational studies including docking and MD simulation revealed several key interactions with the enzyme. The authors demonstrate the non-cytotoxic effect and significant suppression of melanin contents and intracellular tyrosinase activity as well as the expression of tyrosinase. Some comments below:

  1. The compound contains a reactive Michael acceptor. Any possibility of covalent inhibition? Is there any time-dependent inhibition investigated in biochemical assays? The current scaffold could be promiscuous with a deluge of off-targets in the human proteome. The authors should explain the discrepancy between biochemical activity and cellular activity.

Answer: Thank you for your valuable comment. As the reviewer mentioned, the compounds basically have a Michael acceptor moiety. However, the β-carbon of the carbonyl group in the Michael acceptor is linked to a phenyl group. This creates an extended conjugation system and dramatically decreases the ability of the Michael acceptor due to decrease in β positive property at the β-position of the carbonyl group and steric hindrance by the phenyl ring existing at the β -position. In addition, since the carbonyl group is conjugated to the furan ring, this conjugation can also reduce the ability of the Michael acceptor, partially. Previously, we synthesized compounds structurally similar to (E)-1-(furan-2-yl)-(substituted phenyl)prop-2-en-1-one derivatives and tried to convert double bonds to the single bond through a Michael addition reaction using NaBH4. However, it probably doesn’t work for the reasons mentioned above.

  1. The authors investigated MOA by molecular docking. Is that specific interaction that could explain the observed SAR? Without experimental validation like mutation, the docking for the allosteric site is not convincing enough.

Answer: Thank you for your question. Actually we performed enzyme kinetics and docking simulation of compound 8 against mushroom tyrosinase. From the results of kinetic study, we found that compound 8 exerted a mixed-type inhibition of tyrosinase enzyme. This finding regarding the binding ability of compound 8 to both catalytic and allosteric sites confirmed its mixed type inhibition of tyrosinase (Figure 3; Table 2). In other words, compound 8 was found to be able to bind to catalytic and allosteric sites of tyrosinase, which was consistent with its observed mixed mode of inhibition.

  1. The figures look great but typo errors in the manuscript should be carefully checked. Also, Table 1 should be well-organized. Currently, the layout makes it unpleasant for reading. Texts in Figure 3D-I are difficult to see.

Answer: Thank you for your attention. We revised the Figure 3 and Figure legend in the manuscript and noted with red color. And we carefully double-checked the whole manuscript. Please review Figure 3 in the manuscript.

Reviewer 3 Report

(E)-1-(Furan-2-yl)-(substituted phenyl)prop-2-en-1-one derivatives as tyrosinase inhibitors and melanogenesis inhibitors and melanogenesis inhibition: An in vitro and in silico study.

Jung et al.

The present manuscript deals with a new series of inhibitors with potential use in treatments against hyperpigmentation by acting on human tyrosinase as potent inhibitors with low cytotoxicity. In particular, they describe how compound 8 is a much more potent inhibitor than kojic acid, in the same line as some of the most potent chalcones described in this context. The overall presentation, the importance of the study and the quality of the results make this manuscript a very interesting one. However, in my opinion, the authors should make some important modifications before the study could be considered.

One of the most important flaws in this study is the kinetic determination of inhibitory constants. IC50s are calculated by numerical interpolation, inhibition mechanisms determined by visual inspection of intersection points and inhibitory constants obtained from linearisation. While these procedures were acceptable before the advent of potent computers and are still a good way to show your data in figures, nowadays there is little reason for not using non-linear regression methods. Several high-quality software programs are available, some of them open access. The conclusions will probably not be affected, but the quality of the data will improve with increased accuracy and the possibility of applying statistical tools. If the authors cannot get hold of that kind of software, they may calculate IC50s from linearisation, and both IC50s and inhibition constants from several independent determinations (replicates) so that standard error of the mean can accompany those estimations.

Related to this is the misconception of Ki. The authors describe that compound 8 acts as a mixed competitive-uncompetitive inhibitor and they calculate Ki from Dixon plots. They claim those are the values for both the competitive and the uncompetitive inhibition. Ki is indeed the constant related to the competitive inhibition but the uncompetitive inhibition constant is K'i (a.k.a. αKi). If the authors need to use linearisations, they may use re-plots from the lineweaver-burk slopes and intersection points; alternatively, they could use Cornish-Bowden plots to calculate K'i (Cornish-Bowden (1974) Biochem. J. 137: 143-144) using the same data as in the Dixon plots. The importance of this is paramount: it could happen that the values of K'i are very different from Ki, and that would affect the interpretation of the estimated binding energies to the catalytic and allosteric sites (Table 2), among other.

Related to the in silico work, I am no specialist in the field. However, due to their virtual nature, by definition, these in silico determinations cannot be considered a direct measure of what actually happens in nature. At any rate, they are useful because they provide powerful indications of what is the most likely situation. For that reason, the authors should re-write sentences where in silico work are said to prove or confirm the experimental work (e.g. Page 7, line 196, page 9, lines 233 and 239).

With respect to the in vivo assays, I found sub-section 2.8 surprising. The authors say that they wanted to know if "compound 8 could inhibit tyrosinase activity in cells". That was already shown in the previous sub-section. What I think they actually meant was that they wanted to know if compound 8 affected the cellular total activity levels beyond what it is expected from an non-covalent inhibitor. Surprisingly, they find it does. However, the experimental approach to make those measurements are obscure. What was the rationale to do those experiments in the first place? Do they treat the cells with compound 8, collect the cells and wash them prior to make whole cell lysates? How is the washing done? What activity do they measure? Do they use L-Tyr or L-DOPA? Do the authors have an explanation for the phenomenon or, at least, examples of similar situations in the literature?

Regarding language and writing, I strongly recommend the authors to revise the manuscript to correct English language issues (e.g. Page 6, line 174-176), redundancies (e.g. Page 6, lines 177-179 are redundant with lines 169-171 and a similar situation can be found in page 5, between lines 142 and 149), and consistency (e.g. in the legend of Figure 4, (A) and (B) are mentioned before their description, while (C) and (D) are mentioned after). This is by no means a full list.

Also important is the correct presentation of figures and tables. For example, the authors should correct the title of the abscissas in figure 2, panels B and D: instead of "Concentration" probably it should say "[Compound 8]". Similarly, in my copy, table 1 presents some confusing figures for kojic acid IC50s. In Table 2, the manuscript could improve clarity if includes "Allosteric site 1" and "Allosteric site 2" instead of two identical entries as "Allosteric".

Author Response

Reviewer 3

Jung et al.

The present manuscript deals with a new series of inhibitors with potential use in treatments against hyperpigmentation by acting on human tyrosinase as potent inhibitors with low cytotoxicity. In particular, they describe how compound 8 is a much more potent inhibitor than kojic acid, in the same line as some of the most potent chalcones described in this context. The overall presentation, the importance of the study and the quality of the results make this manuscript a very interesting one. However, in my opinion, the authors should make some important modifications before the study could be considered.

  1. One of the most important flaws in this study is the kinetic determination of inhibitory constants. IC50s are calculated by numerical interpolation, inhibition mechanisms determined by visual inspection of intersection points and inhibitory constants obtained from linearisation. While these procedures were acceptable before the advent of potent computers and are still a good way to show your data in figures, nowadays there is little reason for not using non-linear regression methods. Several high-quality software programs are available, some of them open access. The conclusions will probably not be affected, but the quality of the data will improve with increased accuracy and the possibility of applying statistical tools. If the authors cannot get hold of that kind of software, they may calculate IC50s from linearisation, and both IC50s and inhibition constants from several independent determinations (replicates) so that standard error of the mean can accompany those estimations.

Answer: Thank you for your comments. We expressed the tyrosinase inhibitory activity of the tested compounds as an IC50 values (Table 1 in the Results section). These values were calculated from a dose-response inhibition curve using commercially provided EXCEL software program. We present the results for calculated Excel sheet of compound 8 against enzyme tyrosinase using l-tyrosine (A) and l-DOPA (B) as substrates (n=3).

  1. Related to this is the misconception of Ki. The authors describe that compound 8 acts as a mixed competitive-uncompetitive inhibitor and they calculate Ki from Dixon plots. They claim those are the values for both the competitive and the uncompetitive inhibition. Ki is indeed the constant related to the competitive inhibition but the uncompetitive inhibition constant is K'i (a.k.a. αKi). If the authors need to use linearisations, they may use re-plots from the lineweaver-burk slopes and intersection points; alternatively, they could use Cornish-Bowden plots to calculate K'i (Cornish-Bowden (1974) Biochem. J. 137: 143-144) using the same data as in the Dixon plots. The importance of this is paramount: it could happen that the values of K'i are very different from Ki, and that would affect the interpretation of the estimated binding energies to the catalytic and allosteric sites (Table 2), among other.

Answer: Thank you for the detailed comment, we are agreeing with reviewer’s opinion. The compound 8 show a mixed type inhibition for mushroom tyrosinase enzyme. Based on reviewer’s suggestion, we re-calculated the Ki and αKi values from Lineweaver-Burk plots, and revised the subsection of 2.3. enzyme kinetics mechanism study in the manuscript (9 page) and noted with red color.

  1. Related to the in silico work, I am no specialist in the field. However, due to their virtual nature, by definition, these in silico determinations cannot be considered a direct measure of what actually happens in nature. At any rate, they are useful because they provide powerful indications of what is the most likely situation. For that reason, the authors should re-write sentences where in silico work are said to prove or confirm the experimental work (e.g. Page 7, line 196, page 9, lines 233 and 239).

Answer: Thank you for your kind suggestion. We agree with reviewer opinion that in silico study represent a hypothetical result. Therefore, we re-wrote the sentences and noted with red color in the docking results section (12 page, 231 line).

  1. With respect to the in vivo assays, I found sub-section 2.8 surprising. The authors say that they wanted to know if "compound 8 could inhibit tyrosinase activity in cells". That was already shown in the previous sub-section. What I think they actually meant was that they wanted to know if compound 8 affected the cellular total activity levels beyond what it is expected from an non-covalent inhibitor. Surprisingly, they find it does. However, the experimental approach to make those measurements are obscure. What was the rationale to do those experiments in the first place? Do they treat the cells with compound 8, collect the cells and wash them prior to make whole cell lysates? How is the washing done? What activity do they measure? Do they use L-Tyr or L-DOPA? Do the authors have an explanation for the phenomenon or, at least, examples of similar situations in the literature?

Answer: Thank you for your comments. The purpose of cellular tyrosinase inhibitory activity and melanin contents is to investigate effectiveness of skin whitening substances in cell model-based. Therefore, we think these experiments are important. Many researchers are reporting intracellular tyrosinase activity to confirm the anti-tyrosinase activity of active compounds in cellular model. [1-5]. And, methodology for cellular tyrosinase inhibitory activity is described in detail in the Methods and Materials section and subsection of 2.8. Cellular tyrosinase activities and tyrosinase protein levels of compound 8 and noted with red color (27 page). We used l-DOPA as a substrate for cellular tyrosinase inhibition assay and presented a list of references for the assay of cellular tyrosinase inhibition below.

<References>

  1. Kim, Y. J.; No, J. K.; Lee, J. S.; Kim, M. S.; Chung, H. Y., Antimelanogenic activity of 3,4-dihydroxyacetophenone: inhibition of tyrosinase and MITF. Biosci Biotechnol Biochem 2006, 70, (2), 532-4.
  2. Lo, Y. H.; Lin, R. D.; Lin, Y. P.; Liu, Y. L.; Lee, M. H., Active constituents from Sophora japonica exhibiting cellular tyrosinase inhibition in human epidermal melanocytes. J Ethnopharmacol 2009, 124, (3), 625-9.
  3. Kim, J.-K.; Park, K.-T.; Lee, H.-S.; Kim, M.; Lim, Y.-H., Evaluation of the inhibition of mushroom tyrosinase and cellular tyrosinase activities of oxyresveratrol: comparison with mulberroside A. Journal of Enzyme Inhibition and Medicinal Chemistry 2012, 27, (4), 495-503.
  4. Di Petrillo, A.; González-Paramás, A. M.; Era, B.; Medda, R.; Pintus, F.; Santos-Buelga, C.; Fais, A., Tyrosinase inhibition and antioxidant properties of Asphodelus microcarpus extracts. BMC Complement Altern Med 2016, 16, (1), 453.
  5. Ullah, S.; Park, C.; Ikram, M.; Kang, D.; Lee, S.; Yang, J.; Park, Y.; Yoon, S.; Chun, P.; Moon, H. R., Tyrosinase inhibition and anti-melanin generation effect of cinnamamide analogues. Bioorg Chem 2019, 87, 43-55.

  1. Regarding language and writing, I strongly recommend the authors to revise the manuscript to correct English language issues (e.g. Page 6, line 174-176), redundancies (e.g. Page 6, lines 177-179 are redundant with lines 169-171 and a similar situation can be found in page 5, between lines 142 and 149), and consistency (e.g. in the legend of Figure 4, (A) and (B) are mentioned before their description, while (C) and (D) are mentioned after). This is by no means a full list.

Answer: Thank you for your detail comments. We deleted the duplicate sentences and revised the subsection 2.2. Mushroom tyrosinase inhibitory activities of (E)-1-(furan-2-yl)-(substituted phenyl)prop-2-en-1-one derivatives (compounds 1 – 8) in the manuscript. Also, we revised the Figure 4 legend and noted with red color (15 page).

  1. Also important is the correct presentation of figures and tables. For example, the authors should correct the title of the abscissas in figure 2, panels B and D: instead of "Concentration" probably it should say "[Compound 8]". Similarly, in my copy, table 1 presents some confusing figures for kojic acid IC50s. In Table 2, the manuscript could improve clarity if includes "Allosteric site 1" and "Allosteric site 2" instead of two identical entries as "Allosteric".

Answer: Thank you for your comment. First, we revised the Figure 2 and the legend of Figure 2 as Dixon plot is now deleted (Figure 2B, D in the original Manuscript file). And, we revised allosteric sites to allosteric sites 1 and 2 in the subsection 2.4. Molecular docking simulation of compound 8 with mushroom tyrosinase in the Result and Discussion sections.

Round 2

Reviewer 2 Report

Potential covalent MOA needs to be verified by MS instead of looking at the structures. Usually, a probe without good characterization is not very useful. Could be improved in a future study.

Reviewer 3 Report

The authors have anwered all queries I raised. Just a few minor problems were detected:

Page 6, line 187 "tyrosianse" should read "tyrosinase"

Page 9, line 272 "...tyrosinase protein levels of compoud 8" should read "...tyrosinase protein levels in compound 8-treated cells"

Two different issues on Page 13, line 417, the first: "...calculated from Lineweaver-Burk and plot [29, 38]" should read "...calculated from Lineweaver-Burk and re-plots [29, 38]"; the second one: neither reference 29 nor 38 describe the use of re-plots for calculation of Ki and alphaKi. The authors should include a proper reference (e.g. Punekar, NS "Enzymes: Catalysis, Kinetics and Mechanisms" Springer, Singapore, 2018).